

# Transient turbid water mass reduces temperature-induced coral bleaching and mortality in Barbados

Hazel A. Oxenford[1] and Henri Vallès[2]

[1] Centre for Resource Management and Environmental Studies, University of the West Indies, Cave Hill, Barbados

[2] Department of Biological and Chemical Sciences, University of the West Indies, Cave Hill, Barbados

## ABSTRACT

Global warming is seen as one of the greatest threats to the world's coral reefs and, with the continued rise in sea surface temperature predicted into the future, there is a great need for further understanding of how to prevent and address the damaging impacts. This is particularly so for countries whose economies depend heavily on healthy reefs, such as those of the eastern Caribbean. Here, we compare the severity of bleaching and mortality for five dominant coral species at six representative reef sites in Barbados during the two most significant warm-water events ever recorded in the eastern Caribbean, i.e., 2005 and 2010, and describe prevailing island-scale sea water conditions during both events. In so doing, we demonstrate that coral bleaching and subsequent mortality were considerably lower in 2010 than in 2005 for all species, irrespective of site, even though the anomalously warm water temperature profiles were very similar between years. We also show that during the 2010 event, Barbados was engulfed by a transient dark green turbid water mass of riverine origin coming from South America. We suggest that reduced exposure to high solar radiation associated with this transient water mass was the primary contributing factor to the lower bleaching and mortality observed in all corals. We conclude that monitoring these episodic mesoscale oceanographic features might improve risk assessments of southeastern Caribbean reefs to warm-water events in the future.

Corresponding author
Hazel A. Oxenford,
hazel.oxenford@cavehill.uwi.edu

## INTRODUCTION

Coral reefs across the Caribbean have shown significant losses in live coral cover over the last five decades, even though the decline has been spatially and temporally variable (*Pandolfi et al.*, *2003*; *Jackson et al.*, *2014*). These changes have largely been driven by local stressors, especially overfishing and deterioration of water quality (*Rogers*, *1985*; *Pandolfi et al.*, *2003*; *Jackson et al.*, *2014*), which have reduced the innate resilience of these complex communities. As such, we have witnessed increased levels of coral disease (*Aronson & Precht*, *2006*) and substantial shifts to low relief, algal dominated reefs in many locations (*Done*, *1992*; *Hughes*, *1994*; *Alvarez-Filip et al.*, *2009*). These shifts have been accompanied by significant changes in ecosystem structure and function (*Done*, *1992*; *Mumby & Steneck*, *2008*). This pattern of degradation is now being exacerbated by climate variability and

change, which is expected to accelerate coral reef declines across the region over the coming decades (*McWilliams et al.*, *2005*; *Anthony et al.*, *2011*; *Pandolfi et al.*, *2011*).

An evaluation of the mass bleaching response and subsequent high mortality of coral that occurred in Barbados (*Oxenford et al.*, *2008*; *Oxenford, Roach & Brathwaite*, *2010*), in common with other islands throughout the eastern Caribbean (*Wilkinson & Souter*, *2008*; *Eakin et al.*, *2010*), as a result of the anomalous ocean warming event in 2005, demonstrated the substantial damage that can be caused by a single climatic event. It also highlighted the importance of understanding the reef response to subsequent warm water anomalies in order to predict impacts and future risk, especially given that the most recent projections indicate increased incidences and severity of sea surface temperature hotspots in the region (*Van Hooidonk et al.*, *2015*; *Nurse & Charlery*, *2016*).

Here we compare the bleaching and mortality responses of five dominant coral species in Barbados, across three shallow and three deep reefs, observed during the two most significant warm-water events ever recorded in the eastern Caribbean, i.e., 2005 and 2010, and we describe the prevailing sea water conditions at the time of the two events. Our main objective is two-fold: (1) to evaluate how corals responded to consecutive anomalous warming events and; (2) to shed light on the potential role of sea water conditions in mediating the response.

## METHODS

### Benthic surveys

Quantitative benthic surveys were conducted at six reef sites on the semi-exposed southwest and sheltered west coasts of Barbados (3 shallow at <10 m and 3 deep at >15 m), representing the typical range of reef habitats in Barbados (fringing, patch and bank reefs; *Brathwaite, Oxenford & Roach*, *2008*; Fig. 1). Further details (GPS coordinates, exact depths, coral cover) can be found in *Oxenford et al.* (*2008*). For each of the two anomalously warm summers (2005, 2010), surveys were initiated during the period of highest water temperatures (September/October) and were subsequently repeated every four months (in February and June) through the following year. At each site during each survey, five haphazardly placed 20 ×1 m belt transects were surveyed by SCUBA divers. Within each transect all coral colonies were identified to species, counted and classified as either fully bleached (living colony with complete loss of tissue coloration) or not fully bleached (living colony with full or partial tissue coloration). Given the large number of coral species concurrently examined, this index of bleaching was used to minimize the risk of subjective differences in appreciation of partial bleaching levels among divers and within divers over time. We also recorded the percent of a colony's surface that had died recently (i.e., recent dead = exposed white skeleton or light covering with green turf algae).

### The study species

To assess the consistency of the coral response between warm-water events across sites and coral species, we focused on the five most abundant coral species that occurred at all sites, i.e., the species with the five highest average ranks in overall (across surveys) colony abundance across sites. These species together represented more than two thirds (69%) of

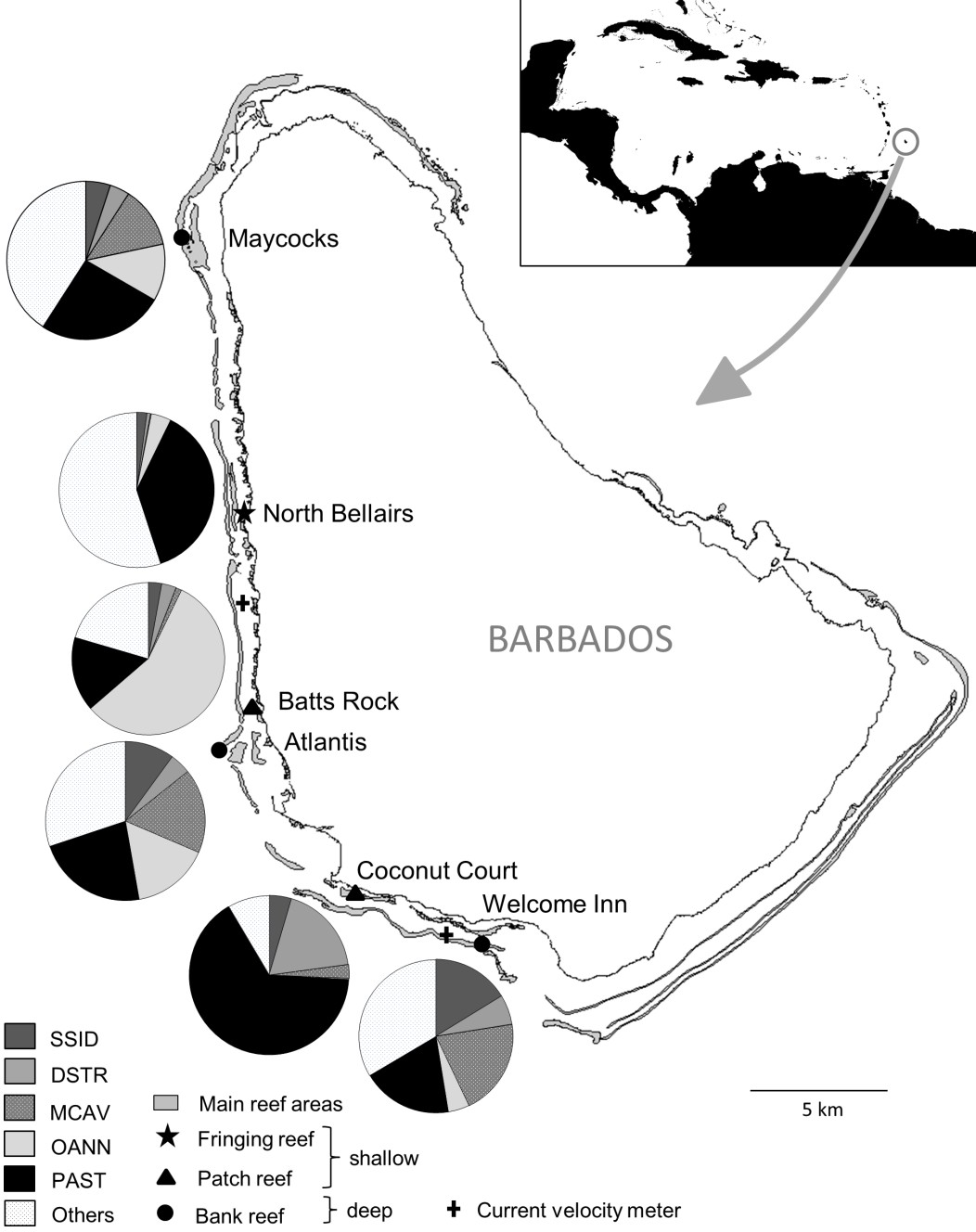

**Figure 1** **Map of Barbados indicating location and characteristics of the six reef sites surveyed, and the two current velocity meters.** The inset shows the location of Barbados in the Caribbean. The pie charts show relative abundance of the five selected dominant coral species at each site. SSID, *Siderastrea siderea*; DSTR, *Diploria strigosa*; MCAV, *Montastraea cavernosa*; OANN, *Orbicella annularis*; PAST, *Porites astreoides*.

all coral colonies surveyed and each species had a cumulative frequency of more than 1,900 colonies examined. The selected corals included a single pioneering 'weedy' species (*Porites astreoides*) and four climax reef-building 'boulder' corals (*Orbicella annularis, Montastraea cavernosa, Diploria strigosa* and *Siderastrea siderea*). Note that *Orbicella annularis* likely

represents a complex of the three very similar species (*O. annularis, O. franksi* and *O. faveolata*) since they could not be reliably distinguished in different environments. The relative importance of the dominant coral species across the different reef sites varied considerably among the three shallow sites, representing different reef habitats, but was similar across the three deep bank reef sites (Fig. 1).

## Sea water environmental data

*In situ* benthic water temperatures were recorded every 4 h using onset® HOBO® Water Temperature Pro v2 Data Loggers placed among the corals at three of the reef sites (Batts Rock, Atlantis and North Bellairs). However, since the temperature profiles from all our loggers were virtually identical (see *Oxenford et al.*, *2008*), we have used the logger data from the site with the longest uninterrupted series (i.e., Batts Rock), for which the data are available from June 2005 onward. Computed mesoscale (50 km grid) degree heating week (DHW) data (*Liu et al.*, *2006*) were accessed from the archived NOAA/NESDIS Coral Reef Watch database at http://coralreefwatch.noaa.gov/ satellite/vs/data_timeseries/vs_ts_Barbados.txt and are available from November 2000 onward. Satellite-derived data on coloured dissolved organic matter index (CDOM), particulate organic carbon concentration (POC) and chlorophyll-*a* concentration were obtained from the MODIS-Aqua 4 km dataset via NASA's Ocean Color Giovanni site at http://gdata1.sci.gsfc.nasa.gov/daac-bin/G3/gui.cgi?instance_id=ocean_month. These data are available from July 2002 onward. We also obtained data on daily current velocity in 2005 and 2010 from the Coastal Zone Management Unit, Government of Barbados, collected by S4 current meters (InterOceans Systems Inc.) at one west coast and one south coast site (Fig. 1). These data are available from 2004 but exhibit large gaps in the time series.

## Analyses

For each coral species at each site, we quantified the bleaching response as the proportion of fully bleached colonies per 20 m$^2$ transect. Furthermore, for each coral species at each site, we quantified coral tissue mortality as percent recently dead tissue per colony per 20 m$^2$ transect.

For each coral species, we assessed differences in response in bleaching and tissue mortality between warm-water events using a median test based on spatially restricted permutations of the data. We used the median test, a simple yet robust non-parametric test (*Sprent & Smeeton*, *2001*), because our bleaching and tissue mortality data exhibited large numbers of zeros (68% and 45% of the overall transect observations, respectively). The testing procedure involved, firstly, assessing the extent to which the median (bleaching and tissue mortality) response differed between events by running the test with the transect data combined across sites to yield an overall chi-square value. Then, we re-ran the test ten thousand times while randomly shuffling the data between both events at each time, but restricted the data shuffling to take place within sites only so as to account for the fact that the data came from different sites. The latter produced a unimodal frequency distribution of pseudo chi-square values ($n = 10,000$) expected under the null hypothesis (i.e., $H_o$: no

difference between events), which accounted for the spatial structure of the data. Finally, we assessed the probability that the overall chi-square value obtained in the first step (with the original data) came from the distribution of pseudo-chi-square values produced by the spatially restricted permutations (*Manly*, *1991*). If the probability of this value was considered too low (i.e., $p < 0.05$), we rejected the null hypothesis and considered the response variable to have differed significantly between events for a given coral species. This testing procedure was repeated for each coral species for each monitoring period. This implied that we ran the test 15 times (5 coral species $\times 3$ monitoring periods) for the bleaching response and ten times (5 coral species $\times 2$ monitoring periods) for the tissue mortality response. To minimize type I errors due to multiple tests, we adjusted $p$-values using sequential Bonferroni corrections (*Holm*, *1979*) for each separate response variable. These analyses were conducted in the R environment (*R Core Team*, *2014*).

## RESULTS

### Bleaching response

The bleaching response to the 2005 warm-water event showed a remarkably similar temporal pattern across all five coral species at the six reefs, with a very high proportion of colonies bleaching during the period of highest water temperatures (September/October), a considerably reduced proportion of bleached corals four months later (February) and almost none by June (Fig. 2 and Table 1).

Likewise the bleaching response to the 2010 warm-water event also showed a similar temporal pattern across all five coral species at the six reefs, with the highest proportion of colonies bleaching in September/October and very low levels of bleaching being recorded during the following year through February and June (Fig. 2 and Table 1).

However, a comparison of the coral bleaching response between the 2005 and 2010 events indicated substantial differences in the magnitude of bleaching (Fig. 2). Indeed, the initial coral bleaching response was considerably less severe in the second event (2010). In all but a single case, the proportion of colonies bleaching for each species at each reef in September/October 2010 was less than half that recorded in the previous event (September/October 2005). Also, the proportion of colonies still bleached in the following February 2011 was in all cases either lower or the same as in February 2006, whilst there was little difference in the low levels of bleaching recorded in the following June of each event. These observations are confirmed by median tests comparing the intensity of bleaching between both events, which show highly significant differences for all five species in September/October ($p = 0.0015$ in all cases), for two species in February (*O. annularis*, $p = 0.0396$; *P. astreoides*, $p = 0.0496$), and just one species in June (*O. annularis*, $p = 0.0360$) (Table 1).

### Tissue mortality response

The coral tissue mortality response that followed the peak period of bleaching also exhibited a similar temporal pattern among the five coral species at the six reefs for the 2005 event, with a relatively low proportion of dead coral tissue being recorded in February 2006 and

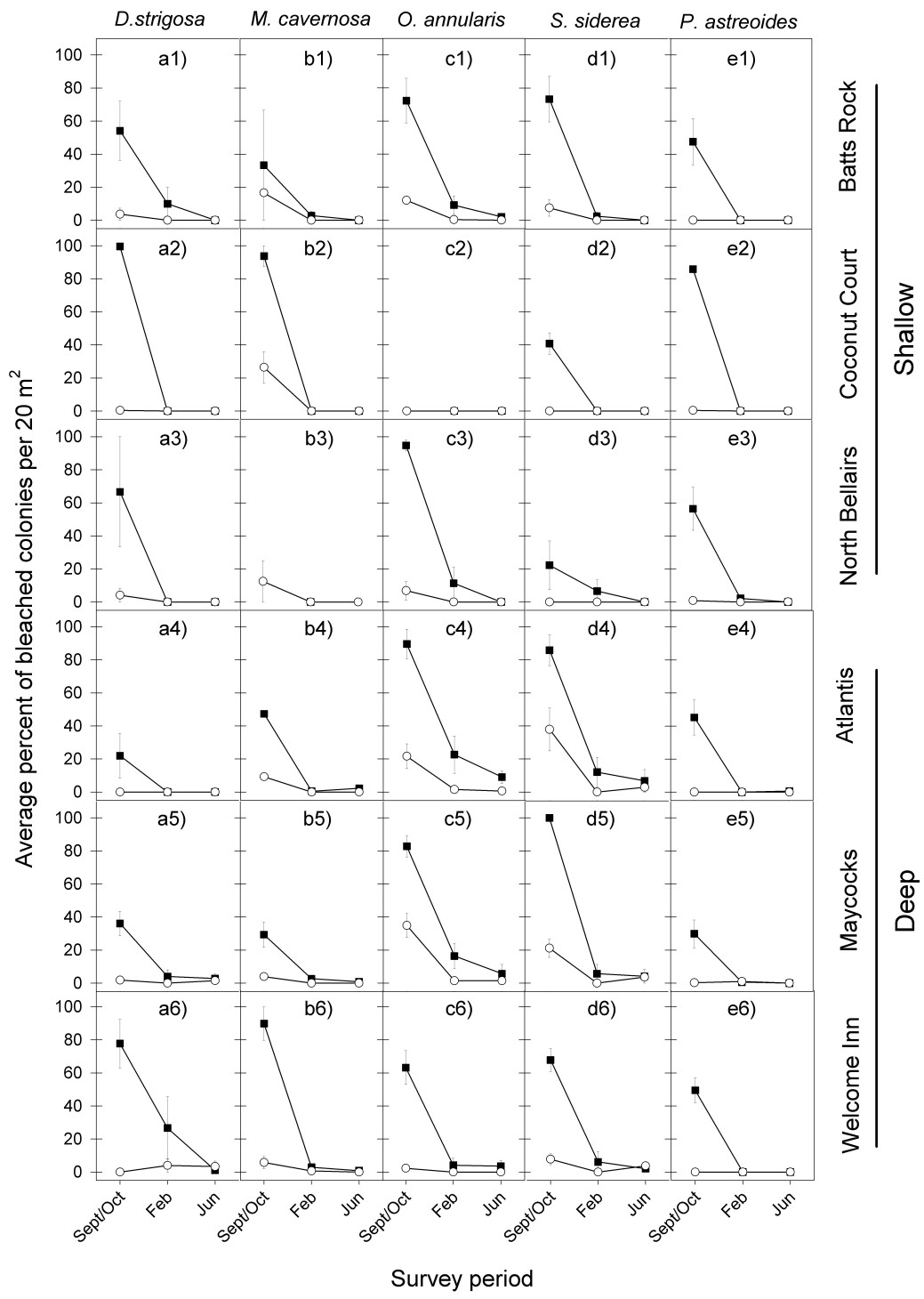

**Figure 2** Average percent of bleached colonies during peak bleaching (September/October) and the following post-bleaching (February and June) survey periods in 2005–2006 (black squares) and 2010–2011 (white circles) for the five selected dominant coral species (A–E) across the six study sites (1–6). Error bars denote 1 standard error. Lack of symbols in b3 panel indicates that no *M. cavernosa* colonies were sampled in Sept/Oct 2005 and in June 2006 at the North Bellairs site; and in c2 panel no *O. annularis* colonies were sampled in Sept/Oct 2005 at the Coconut Court site.

**Table 1 Summary statistics comparing the overall intensity of coral bleaching between 2005/2006 and 2010/2011 during the same survey months.** The mean percent of bleached colonies across the six sites (see Fig. 2 for individual site values) and associated standard deviation (in brackets) is given for each survey month and year. The results of the global (incorporating all sites) median tests comparing the intensity of bleaching between years are also shown (chi-square statistics and associated $p$ values). For these tests, the transect data (as percent of bleached colonies in a given transect) were used as replicates. $P$ values were obtained by permuting transect data within sites (10,000 permutations) and were subsequently adjusted to account for multiple tests (Holm, 1979).

| Species | September/October | | | |
|---|---|---|---|---|
| | **2005** | **2010** | **Chi-square** | **$p$ value** |
| *Diploria strigosa* | **59.4 (28.2)** | **1.7 (1.9)** | **20.3** | **0.0015** |
| *Montastraea cavernosa* | **58.7 (30.9)** | **12.4 (8.2)** | **9.4** | **0.0015** |
| *Orbicella annularis* | **80.6 (12.8)** | **13.0 (13.2)** | **30.1** | **0.0015** |
| *Siderastrea siderea* | **65.0 (28.8)** | **12.4 (14.7)** | **22.3** | **0.0015** |
| *Porites astreoides* | **52.4 (18.6)** | **0.2 (0.3)** | **56.1** | **0.0015** |

| Species | February | | | |
|---|---|---|---|---|
| | **2006** | **2011** | **Chi-square** | **$p$ value** |
| *Diploria strigosa* | 6.8 (10.5) | 0.7 (1.6) | 1.8 | 0.5691 |
| *Montastraea cavernosa* | 1.5 (1.5) | 0.1 (0.3) | 2.6 | 0.5691 |
| *Orbicella annularis* | **10.6 (8.2)** | **0.5 (0.7)** | **7.4** | **0.0396** |
| *Siderastrea siderea* | **5.5 (4.1)** | **0.0 (0.0)** | **6.5** | **0.0496** |
| *Porites astreoides* | 0.5 (0.9) | 0.2 (0.4) | 1.7 | 0.6048 |

| Species | June | | | |
|---|---|---|---|---|
| | **2006** | **2011** | **Chi-square** | **$p$ value** |
| *Diploria strigosa* | 0.6 (1.1) | 0.8 (1.4) | 0.0 | 1.0000 |
| *Montastraea cavernosa* | 0.8 (0.9) | 0.0 (0.0) | 2.3 | 0.5691 |
| *Orbicella annularis* | **3.4 (3.5)** | **0.3 (0.6)** | **7.2** | **0.0360** |
| *Siderastrea siderea* | 2.2 (2.8) | 1.7 (1.9) | 0.1 | 1.0000 |
| *Porites astreoides* | 0.1 (0.3) | 0.0 (0.1) | 0.0 | 1.0000 |

Notes.
Bold font indicates statistical significance at the 0.05 level.

a considerably higher proportion of dead coral tissue being recorded in June 2006 (Fig. 3 and Table 2).

There was also a high degree of similarity in temporal patterns of coral tissue mortality among most species at most reefs following the 2010 warm-water event, with very low proportions of dead coral tissue in both February and June 2011. The only exception was *S. siderea* at two of the six reef sites (Fig. 3 and Table 2).

In line with the finding that coral bleaching was considerably less severe in the second warm-water event (2010) compared with the first one (2005) for all five corals, so too was the post-bleaching tissue mortality response of these five species. In February 2011, the average percent of recently dead coral tissue per colony for each species was the same or slightly less than in February 2006, but by June 2011 it was substantially lower than in June 2006, with the exception of *S. siderea* at the Coconut Court reef site (Fig. 3). Once again

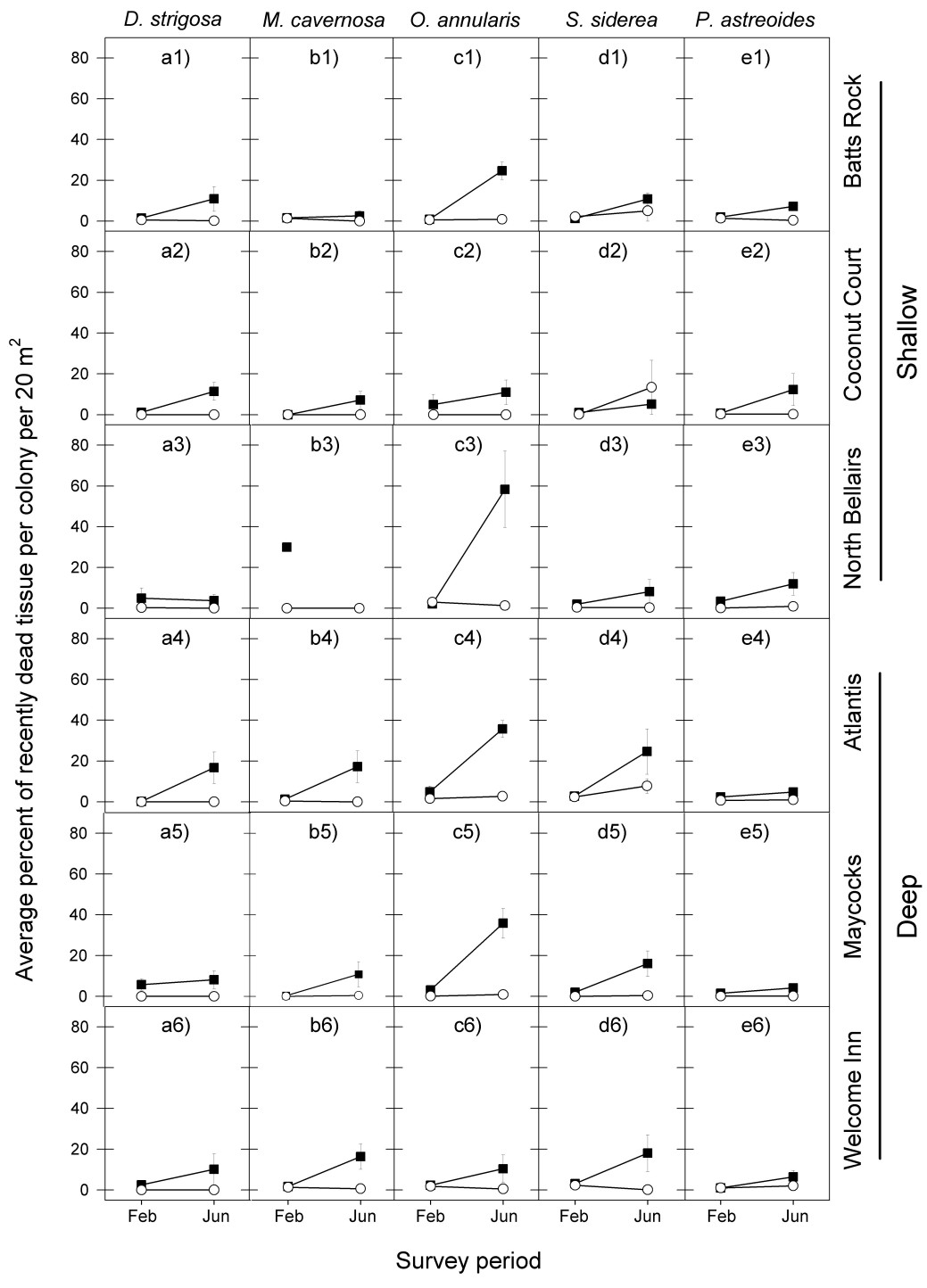

**Figure 3   Average percent of recently dead coral tissue per colony during the post-bleaching survey periods (February and June) in 2006 (black squares) and 2011 (white circles) for the five selected dominant coral species (A–E) across the six study sites (1–6).** Error bars denote 1 standard error. Missing symbol in b3 panel indicates that no *M. cavernosa* colonies were sampled at the North Bellairs site in June 2006.

**Table 2 Summary statistics comparing the intensity of post-bleaching coral mortality (across all sites) between 2005/2006 and 2010/2011 using the same survey months.** The mean percent of recently dead tissue per colony across the six sites (see Fig. 3 for individual site values) and associated standard deviation (in brackets) is given for each survey month and year. The results of the global (incorporating all sites) median tests comparing the intensity of tissue mortality between years are also shown (chi-square statistics and associated *p* values). For these tests, the transect data (as mean percent of recently dead coral tissue per colony in a given transect) were used as replicates. *P* values were obtained by permuting transect data within sites (10,000 permutations) and were subsequently adjusted to account for multiple tests (Holm, 1979).

| Species | February | | | |
|---|---|---|---|---|
| | 2006 | 2011 | Chi-square | *p* value |
| *Diploria strigosa* | **2.7 (2.2)** | **1.2 (1.3)** | **6.9** | **0.0275** |
| *Montastraea cavernosa* | 5.8 (11.9) | 0.5 (0.6) | 1.3 | 0.6840 |
| *Orbicella annularis* | 3.1 (1.7) | 1.2 (1.2) | 1.3 | 0.6840 |
| *Siderastrea siderea* | 2.0 (0.8) | 1.2 (1.2) | 0.8 | 0.6840 |
| *Porites astreoides* | 1.8 (1.0) | 0.6 (0.5) | 0.6 | 0.6840 |

| Species | June | | | |
|---|---|---|---|---|
| | 2006 | 2011 | Chi-square | *p* value |
| *Diploria strigosa* | **10.2 (4.3)** | **0.8 (1.4)** | **22.9** | **0.0010** |
| *Montastraea cavernosa* | **10.8 (6.2)** | **0.2 (0.3)** | **19.5** | **0.0010** |
| *Orbicella annularis* | **29.4 (18.1)** | **1.0 (0.9)** | **24.3** | **0.0010** |
| *Siderastrea siderea* | **13.8 (7.2)** | **4.5 (5.4)** | **22.3** | **0.0010** |
| *Porites astreoides* | **7.8 (3.5)** | **0.8 (0.7)** | **16.9** | **0.0010** |

**Notes.**
Bold font indicates statistical significance at the 0.05 level.

these observations are borne out by the median tests showing no significant differences between the events in the intensity of post-bleaching mortality for four of the five species in February (*D. strigosa*, $p = 0.0275$; all other species, $p = 0.6840$), but significant differences for all species by June (Table 2).

## Sea water environmental conditions

The *in situ* water temperature profiles, unprecedented heating stress (satellite-derived degree heating week (DHW) indices) and satellite derived sea surface temperature (SST) profiles experienced by the reefs during both mass bleaching events in 2005 and 2010 are shown in Figs. 4 and 5A. These profiles indicate similar timing, duration and intensity of the warm-water events in both years, although the 2010 event was slightly more severe with an October mean peak of >10 DHW in 2005 and >11 DHW in 2010 (Fig. 4). In contrast to the similar temperature profiles, satellite-derived sea surface indices negatively associated with water clarity, i.e., coloured dissolved organic matter index (CDOM), particulate organic carbon concentration (POC) and chlorophyll-*a* concentration, differed dramatically between events in the water surrounding Barbados reefs, with unusually high levels (well above the monthly median values for the 2003–2010 period) of CDOM, POC and chlorophyll-*a* during most of the 2010 warming event (Fig. 5, Figs. S1 and S2). This corroborates with *in situ* observations of dark green, turbid water over Barbados

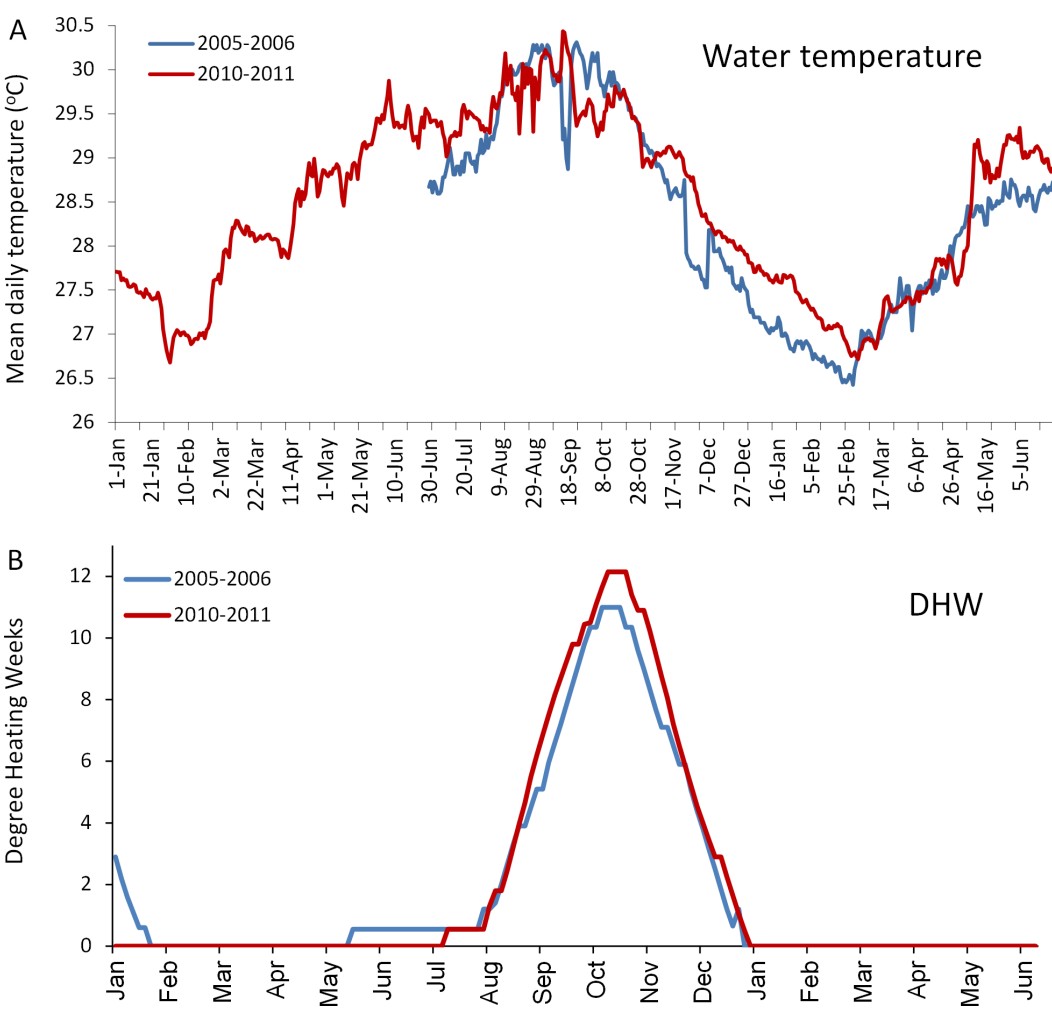

**Figure 4** **Comparison of the temperatures experienced by coral reefs in Barbados during the 2005/2006 (blue line) and 2010/2011 (red line) mass bleaching events.** (A) shows mean daily *in situ* water temperatures (data are only available from June 2005); (B) shows satellite-derived accumulated heating stress as bi-weekly computed degree heating weeks (http://coralreefwatch.noaa.gov/satellite/vs/ data_timeseries/vs_ts_Barbados.txt).

reefs during the summer and fall of 2010 (Pilots log, Atlantis Submarines Barbados Inc.; member's personal record, Barbados Open Water Swimming Club; anecdotal reports from dive operators; and personal observation, see Fig. 6). These satellite-derived data also suggest that the water experienced by the reefs throughout most of the 2005 event was clearer than usual (Fig. 5).

*In situ* data on daily current velocity during the 2005 and 2010 warm-water events at the west and south coast sites were sporadic with only 13 days of overlapping records from both locations in both years, precluding any meaningful statistical comparisons between sites and years (Fig. S3). Visual examination of the time series suggests a lack of temporal consistency in current velocity between these two sites. However, it does reveal evidence of sustained higher current velocity (up to 18.2 cm/s) for the period August–October at the west coast site in 2010 compared with 2005 (the latter ranging from around 3–9 cm/s)

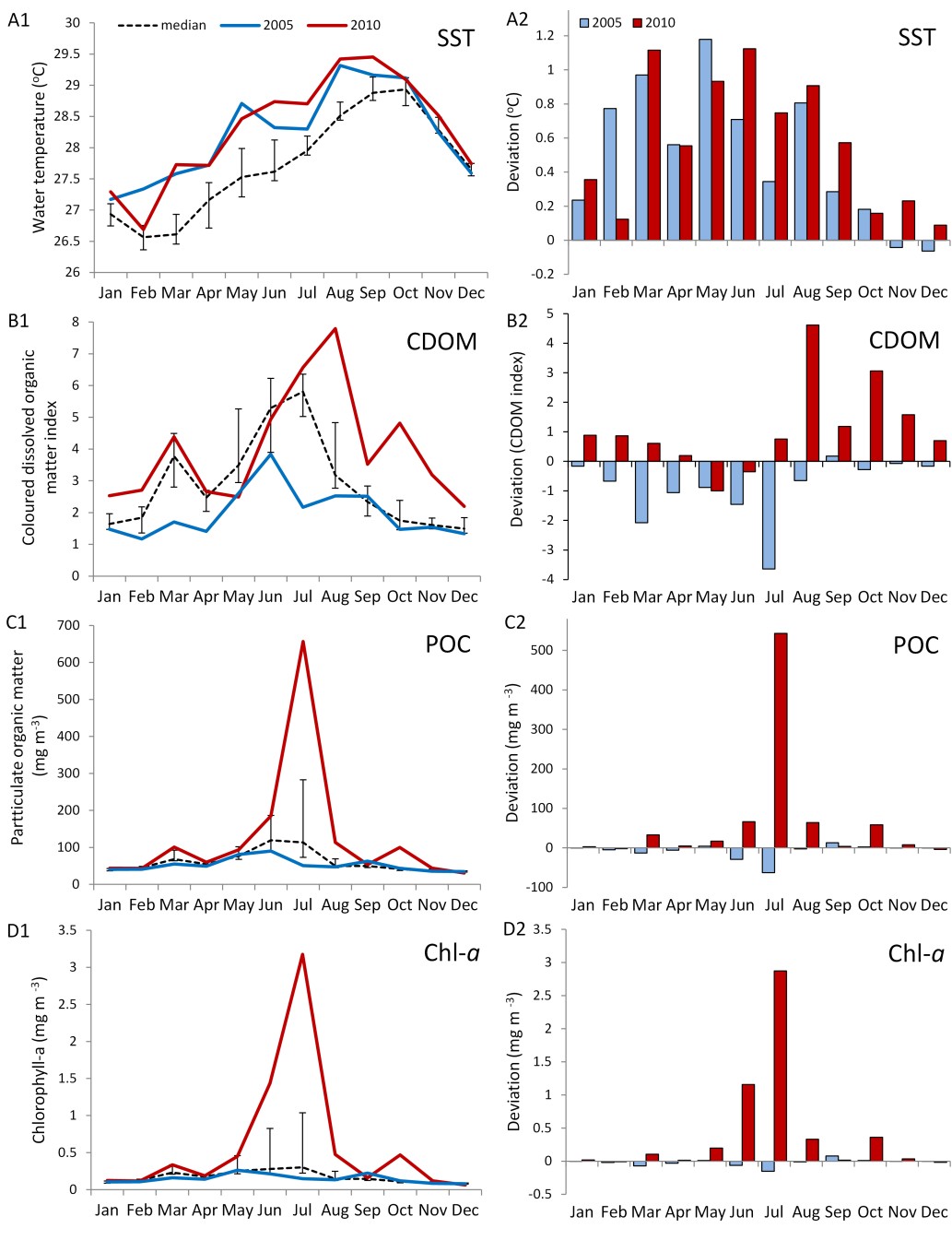

**Figure 5** **Comparison of environmental conditions experienced by coral reefs in Barbados during the 2005 (blue) and 2010 (red) warm water events.** Data are from MODIS-Aqua satellite 4 km computed monthly means for (A) sea surface temperature (SST), (B) an index of coloured dissolved organic matter (CDOM), (C) particulate organic carbon concentration (POC), and (D) chlorophyll-*a* concentration in water surrounding Barbados (lat. 12.889–13.506°N, long. 59.900–59.237°W) (http://gdata1.sci.gsfc.nasa.gov/daac-bin/G3/gui.cgi?instance_id=ocean_month). A1–D1 show monthly means for both years and median values over an 8-year period (2003–2010) (black dashed line with bars indicating 1st and 3rd quartiles). A2–D2 show a comparison of the deviation from the median values for 2005 and 2010. (Maps of time-averaged data for CDOM, POC and Chl-*a*, are given in Figs. S1 and S2).

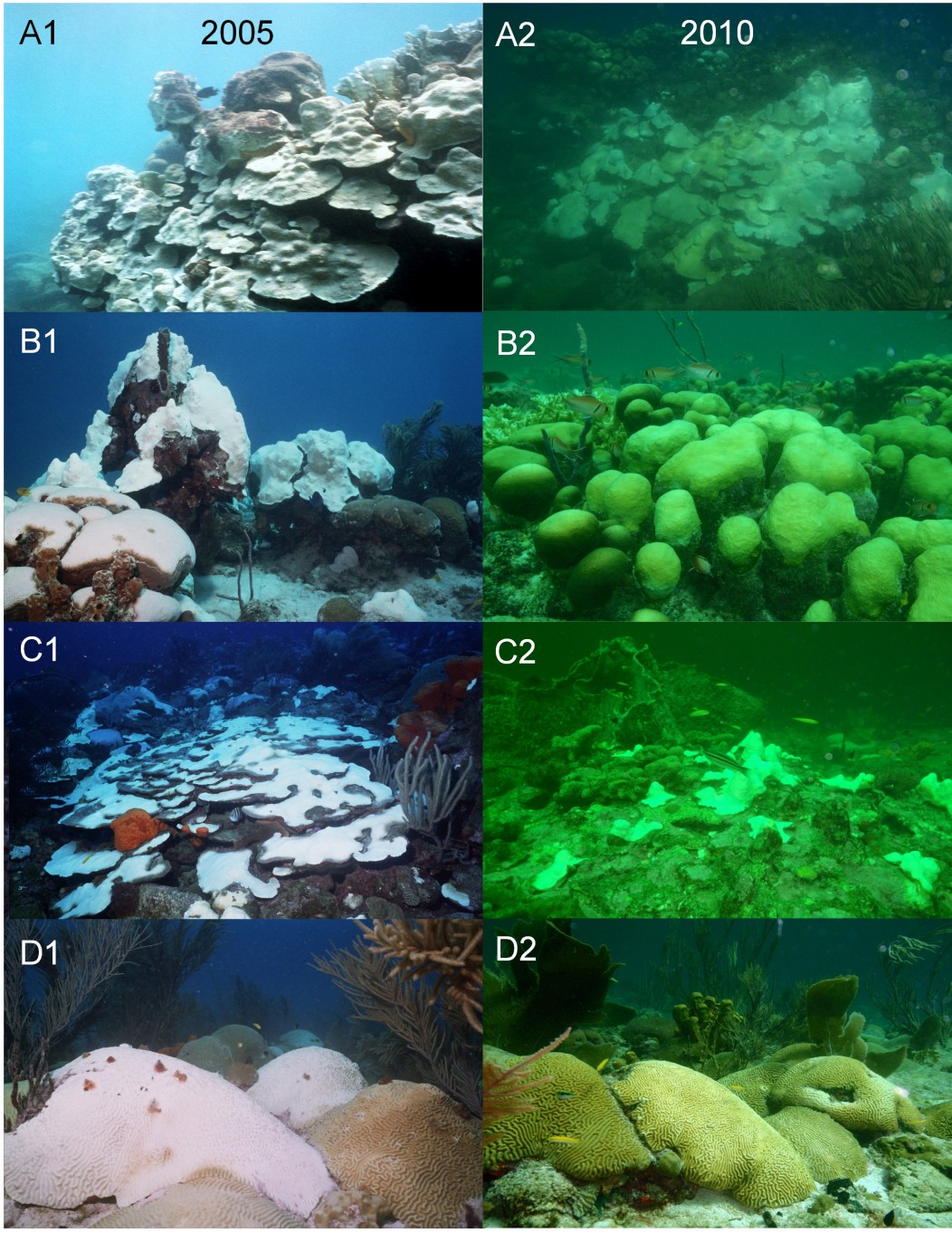

**Figure 6** Photographs taken with the same camera, between 10 am and 3 pm during the coral bleaching surveys in October of the two warm-water events in 2005 (A1–D1) and 2010 (A2–D2) on Barbados reefs, showing the marked difference in water colour and light penetration. (A) North Bellairs fringing reef; (B) Batts Rock patch reef; (C) Atlantis bank reef; (D) Coconut Court patch reef.

(Fig. S3). There is no evidence of concomitant increases in current velocity at the south coast site, although the large data gaps at this site preclude any rigorous conclusion.

## DISCUSSION

In this study we compared the bleaching and post-bleaching mortality responses of the dominant coral species in Barbados to two successive anomalous ocean warming events, i.e., 2005 and 2010, and considered a number of environmental factors that may mitigate the impacts of future acute warming events, the latter being expected to increase in frequency and even become annual in Barbados by 2045–50 (*Van Hooidonk et al.*, *2015*).

We found considerable differences between the two severe warm-water events in the bleaching and mortality response of the dominant coral species; a result that was consistent across all species and reef habitats. In general, the proportion of coral colonies bleaching and the loss of live coral tissue was significantly less in the second event (2010), even though the timing, intensity and duration of both warm-water events were similar or even slightly more severe in 2010.

Numerous studies have demonstrated that the mass bleaching and mortality responses of corals to critically high water temperatures vary, somewhat predictably, according to a number of different parameters. One important parameter is the magnitude of thermal stress (*Marshall & Baird*, *2000*; *McClanahan et al.*, *2007*; *Kleypas, Danabasoglu & Lough*, *2008*). This is commonly measured using an index of accumulated heat stress, for example the degree heating week (DHW) index used by NOAA (http://coralreefwatch.noaa.gov/), which assimilates heating stress over the previous 12-week period. It is generally agreed that mass coral bleaching is evoked when DHW $= 4\,°C$ heating weeks, and mass mortality will occur when DHW $\geq 8\,°C$ (*Liu et al.*, *2006*). The higher the heating stress above these thresholds, the more severe the response (*Eakin et al.*, *2010*). In this case however, the *in situ* water temperature and the satellite-derived DHW profiles for each event were closely matched and exceeded 10 DHW in both years (even being slightly higher in 2010), negating differences in magnitude of heating stress as a plausible explanation for the observed differences in the general response of all five dominant coral species.

Other parameters influencing susceptibility of corals to heating stress (see *West & Salm*, *2003*; *Baker, Glynn & Riegl*, *2008* for reviews) include *inter alia* (1) innate differences among coral species (*Loya et al.*, *2001*; *Baird et al.*, *2009*; *Wagner, Kramer & Van Woesik*, *2010*; *Barshis et al.*, *2013*) as well as differences in the zooxanthellae community they typically host (*Berkelmans & Van Oppen*, *2006*; *LaJeunesse et al.*, *2009*; *Baums, Devlin-Durante & Lajeunesse*, *2014*), (2) differences in colony size (*Brandt*, *2009*; *Wagner, Kramer & Van Woesik*, *2010*); and (3) differences in reef habitats (*Marshall & Baird*, *2000*; *Chollett, Enriquez & Mumby*, *2014*). Again, although these factors may partially explain some of the observed variation in response among coral species or among reefs in any given event, they cannot be driving the differences we observed between the two events, since we were monitoring the same group of species over the same depth ranges and habitats (same reef sites) over time and had large sample sizes covering a wide range of colony sizes. Furthermore, although there was some variation in the severity of the bleaching and

mortality responses among species in any given event (as expected), the relative difference in response between the two events was consistent across all species with just one exception (one species at just one of six reef sites in one of three survey periods), and the general response patterns were consistent across all five dominant species.

Bleaching and post-bleaching mortality responses may also be exacerbated by the presence of other chronic local stressors causing synergistic effects (see *Wagner, Kramer & Van Woesik*, *2010*; *Van Woesik et al.*, *2012*; *Ateweberhan et al.*, *2013* and references therein). The main chronic local stressors on Barbados reefs are eutrophication from land-based activities (*Bell & Tomascik*, *1994*) and overfishing (*Government of Barbados*, *2002*; *McConney*, *2011*), both of which are known to vary among reefs (especially between nearshore shallow and offshore deep reefs). Spatial variability in such stressors may indeed explain observed differences in the severity of responses of single coral species among different reefs in any given event. However, these chronic local stressors have not changed dramatically over the 5-year period between events and are therefore extremely unlikely to be responsible for the difference in responses observed between the 2005 warm-water event and the 2010 one. For example, reef fish landings and number of active reef fishing boats has remained stable in Barbados over the period 1997–2010 (*Schuhmann et al.*, *2011*) and water quality data (phosphate and ammonia concentrations) collected at three of our reef sites on the south (Coconut Court) and west (Atlantis and North Bellairs) coasts by the Government's Coastal Zone Management Unit (CZMU) show no consistent trends in eutrophication over the period 2006–2010 (CZMU, 2016, unpublished data).

Alternatively, it is possible that the bleaching responses we observed in 2010 were mediated by previous coral experience with warm-water events (*Oliver & Palumbi*, *2011*; *Guest et al.*, *2012*). This 'acclimatization' may occur through a complex variety of mechanisms that include physiological alterations in one or more of the individual partners of the coral holobiont symbioses (coral host, algal symbionts, microbial associates) or changes in the identities or composition of the holobiont's algal and microbial communities (see *Baker, Glynn & Riegl*, *2008* for review). A study of eight coral species (including all five studied here) on Barbados' reefs during and up to two years after the 2005/2006 bleaching event did indicate algal symbiont 'shuffling' and significant increases in the proportion of heat-resistant *Symbiodinium trenchi* hosted by some species in response to thermal stress (*LaJeunesse et al.*, *2009*). However, not all coral species showed significant shuffling of their algal symbionts (namely three of our five species: *S. siderea*, *Diploria* spp. and *P. astreoides*) and those that did increase the proportion of heat resistant symbionts, were found to revert to their original symbiont community within two years of the event (*LaJeunesse et al.*, *2009*). Preliminary studies have reported some level of acclimation to heat stress in a single coral species through other physiological mechanisms, namely gene expression (*Barshis et al.*, *2013*; *Palumbi et al.*, *2014*). However, the corals studied were from an environment in which they were habitually exposed to large fluctuations in temperature associated with the daily tidal cycle, and therefore the results are unlikely to be transferable to corals in our own study which are periodically exposed (e.g., at multiple year intervals) to high temperatures for many weeks at a time. Given this and the fact that the observed symbiont shuffling in our coral species was only temporary (less than 2 years; *LaJeunesse et al.*, *2009*),

together with the fact that acclimatization or even adaptation is likely to be highly variable among coral taxa, it seems unlikely that acquired heat tolerance could adequately explain why all five dominant coral species examined were considerably less affected by the 2010 warming event, which occurred five years after the first exposure.

We therefore believe that the explanation must lie with one or more factors mediating the ambient environment experienced by all coral species. For example, since a mass bleaching response in corals is elicited through a combination of heating stress and irradiance (*Iglesias-Prieto et al.*, *1992*) (see *Fitt et al.*, *2001*; *Baker, Glynn & Riegl*, *2008* for reviews), a reduction in irradiance could decrease the impact of heating stress (*Glynn & D'Croz*, *1990*). High solar radiation, particularly photosynthetically active radiation (PAR), disrupts biochemical pathways and damages the photosynthetic apparatus of the symbiotic zooxanthellae, resulting in the production of toxins (oxygen radicals and other detrimental photosynthetic byproducts), which ultimately result in the expulsion of the symbionts from host tissue (*Baker, Glynn & Riegl*, *2008*). Environmental factors that allow for improved flushing of these toxic metabolites that have been found to mitigate bleaching and improve recovery rates include high water current flow under laboratory conditions (*Nakamura & Van Woesik*, *2001*; *Nakamura, Yamasaki & Van Woesik*, *2003*) and associated with upwelling or wind exposure (*Bayraktarov et al.*, *2013*). Environmental factors reducing solar radiation that have been reported to reduce the severity of coral bleaching include shading by high islands (*West & Salm*, *2003*) or in cryptic habitats (*Mumby*, *1999*), cloud cover (*Mumby et al.*, *2001*), increased turbidity (*Jokiel & Brown*, *2004*; *Otis et al.*, *2004*; *Van Woesik et al.*, *2012*) and presence of high levels of aerosols in the atmosphere (*Gill et al.*, *2006*), all of which block or scatter and attenuate radiation. In addition, ambient conditions leading to an increase in heterotrophic feeding by the coral host could prevent starvation and subsequent mortality in bleached corals (*Hughes & Grottoli*, *2013*) and might play an important role in preventing the onset of temperature-induced bleaching (*Wooldridge*, *2014*).

In this case, Barbados and other islands in the eastern Caribbean (see mention by *Alemu & Clement*, *2014* and Figs. S1 and S2) experienced a significant dark green water mass throughout the summer and fall of 2010. Such green water masses may be experienced in Barbados several times a year due to the episodic passage of North Brazil Current (NBC) rings transporting Amazon River water into the southeastern Caribbean (*Fratantoni & Glickson*, *2002*). The hydrological effects of these water masses of South American riverine origin on Barbados have been well documented *in situ* (Table S1). Of particular relevance to this discussion these effects include, changes in water colour associated with a reduction in light penetration, consistent with our own *in situ* observations (Fig. 6), increases in plankton concentration (*Borstad*, *1979*; *Kidd & Sander*, *1979*), as well as increases in current velocity (*Cowen & Castro*, *1994*; *Stansfield et al.*, *1995*; *Paris et al.*, *2002*), with the magnitude of such current velocity increases depending on the angle at which these NBC rings impinge upon the island (*Cowen et al.*, *2003*).

We therefore suggest that the coincidental passage of an NBC ring during the warm-water event of 2010 was responsible for reducing overall stress on corals in that year. We posit that the high levels of CDOM, POC and chlorophyll-*a* observed island-wide, at the time of peak warm temperatures in 2010 (Fig. S1) associated with the passage of the

NBC ring, was the principle driver in reducing overall stress on corals across all sites by decreasing their exposure to high solar radiation, thereby ultimately reducing bleaching. We also recognize additional factors associated with the NBC ring that could further contribute to mitigating the impact of a warm-water event including (1) an increase in heterotrophic feeding efficiency by corals in the nutrient rich water, and (2) an increase in toxic metabolite flushing through increases in current velocity. In this case, however, we believe that these additional factors do not adequately account for the consistent patterns across species and sites that we observed. Firstly, heterotrophic feeding plasticity is highly species-specific (*Hughes & Grottoli*, *2013*; *Levas et al.*, *2016*). Secondly, current velocities were mostly well below those reported to benefit corals exposed to high temperatures or recovering from bleaching (i.e., sustained water flow ≥20 cm/s; *Nakamura & Van Woesik*, *2001*; *Nakamura, Yamasaki & Van Woesik*, *2003*) and increases in current velocity in 2010 were not apparent at both west and south coast sites, suggesting the existence of substantial spatial heterogeneity in current structure across Barbados likely resulting from the interaction of the NBC rings with Barbados' bathymetry (*Paris et al.*, *2002*).

## CONCLUSION

In summary, our study supports the idea that temperature-induced coral bleaching and mortality can be moderated by temperature-independent mesoscale hydrological features, as has been suggested in other field studies (*Otis et al.*, *2004*; *Van Woesik et al.*, *2012*; *Bayraktarov et al.*, *2013*; *Wall et al.*, *2014*).

In Barbados, such mesoscale hydrological features are often associated with the passage of NBC rings (Table S1), thus contributing to the global list of oceanographic phenomena potentially mediating coral responses to climate change. Importantly, our increasingly sophisticated ability to monitor the NBC ring trajectories from satellites (*Fratantoni & Glickson*, *2002*) provides a tangible opportunity to increase our capacity to predict the severity of temperature-induced mass coral bleaching and mortality events in Barbados as well as in other southeastern Caribbean islands subject to the same oceanographic phenomena. Overall, better prediction should lead to better policy responses, and repeated coincidence of transient turbid water masses or other mitigating oceanographic features with warm-water events could help buy time for local managers struggling to reduce local stressors to improve reef resilience in the face of climate change.

## ACKNOWLEDGEMENTS

We gratefully acknowledge the SCUBA diving assistance and field data collected by colleagues from: the Coastal Zone Management Unit of the Government of Barbados—A Brathwaite, R Roach and F Hinds; and the University of the West Indies—R Goodridge, K Baldwin and C Finney.

### Funding

The authors received no funding for this work.

### Competing Interests

The authors declare there are no competing interests.

### Author Contributions

- Hazel A. Oxenford and Henri Vallès conceived and designed the experiments, performed the experiments, analyzed the data, wrote the paper, prepared figures and/or tables, reviewed drafts of the paper.

### Data Availability

The raw data has been supplied as Data S1 containing all mortality and bleaching data.

### Supplemental Information

Supplemental information for this article can be found online at http://dx.doi.org/10.7717/peerj.2118#supplemental-information.

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
