# Peer review of "Transient turbid water mass reduces temperature-induced coral bleaching and mortality in Barbados"

_PeerJ, doi:10.7717/peerj.2118_

## Round 0.1 · original submission · Major Revisions

I have now received comments from two reviewers. While both reviewers were generally positive in their comments on your paper, reviewer #2 in particular has expressed concerns about the definition of bleaching (all or none), and also requested more information about the sites examined. I agree with these comments, as well as the other helpful comments offered by both reviewers. Finally, both reviewers have discussed your statistical analyses - please review and consider these comments in detail as well. For these reasons, my decision is 'major revision'.

·

Basic reporting

This is a very interesting study based on a unique and comprehensive dataset.
The article is well written and uses relevant references. It should certainly be published.

Experimental design

Well done.

Validity of the findings

Well written and structured. Here are some improvement suggestions:
1) More informative than Fig. 1 would be a presentation that shows for the different study locations how much percent of all hard coral species have been covered by the 5 selected species.
2) Tables 2 and 3 are slightly confusing in my opion. It would be better to clearly describe in text and table or figure how much more the 5 selected species were suceptible for bleaching and subsequent mortality in 2005/2006 compared to 2010/2011.
3) Fig. 5 and 6 present the key data of this manuscript. So, in my opinion they are better located at the beginning of the results section followed by a clear summary figure or table.
4) To disentangle the relative effects of SST, DOM, POC, and chlorophyll a, on bleaching suceptibility and subsequent mortality, it would be great to do some kind of multivariate analysis. In this context, suface current velocity may be another key factor that could be derived form the satellite data to include this factor in the analysis as we know from several studies that water currents may mitigate bleaching.
5) There are three shallow and three deeper locations used for the surveys, but I could not see a clear spatial comparison of bleaching suceptibility between shallow and deep locations. It may be worth to introduce another figure that explicitly targes this interesting comparison.

·

Basic reporting

This study investigated the extent of coral bleaching between the major bleaching event in 2005 and 2010. Coral bleaching was found to be less damaging in 2010 which coincided with the occurrence of turbid water masses. The plankton-rich turbid water potentially decreased the effect of light intensity-induced coral bleaching while delivering additional sources for heterotrophy of heat-stressed corals. The manuscript is well written but needs major restructuring with focus on the scientific question on how and why turbid water masses prevented Barbados’s corals from bleaching. Major flaws are the missing site description (e.g. lack of maps indicating the sampling locations), including of dispensable information which is not necessary for understanding the major results (e.g. Table 1), the lack of justification and referencing of the statistical methods used for analyses, the graphical representation, as well as the repeatedly deviating focus from effects of turbid waters towards e.g. necessity for management actions of Barbados. Yet, in time of increasing climate change and the need for further understanding on how to prevent or address the deteriorating consequences for coral reefs, this study would have global, rather than local significance if the authors would focus their argumentation towards the fact that green turbid water masses can offer coral reefs a refuge habitat against coral bleaching through ameliorating the damaging effects of strong solar radiation coinciding with increased heat stress. I recommend a major revision of the manuscript and would be happy to recommend it for publication after a significant improvement.

Specific comments:
The title is too long and not really catchy; it is very vague. I suggest shorten it up to: 'Turbid water reduces temperature-induced coral bleaching and mortality in Barbados' or similar.

I couldn’t find any figures indicating the study sites. Please add a map with all sampling locations as well as a short description of the study sites other than referring to the authors’ previous publication.
Line 67: I suggest, describing the study site briefly in one or two sentences to avoid that the reader needs to search for Oxenford et al 2008 to get any further clue on the environmental settings of the location. You can still refer to this study for more details.

Experimental design

I don’t agree with the definition of fully bleached and non-fully bleached coral. A coral, which is partially bleached is still bleached and will be disadvantaged compared to such which have no partially bleached tissue. The categories should be rather: bleached (fully or partial), pale, or normal. Another possibility could be to say that if more than 50% of the tissue is bleached, the respective coral accounts as bleached. Your definition is not conservative since it is biased towards non-bleached corals, i.e. you might be underestimating the impact of bleaching if accounting every partially-bleached coral as non-bleached.

Line 81 – 83: Your post-hoc suggestion on how the dominant corals for further investigation have been selected appears somewhat weary and arbitrary. I would rather agree with an argument that corals x, y, and z were selected because together they represented over 90% of the corals at the site.

Line 90-91: Were the loggers placed at the respective reef depth? How many loggers were placed and where exactly?

Validity of the findings

Line 103-121: I am not particularly convinced with the statistically methods used within the manuscript. Also, references on the statistical methods are fully missing (except for the Bonferroni corrections). The statistical analyses would be more convincing if other studies could be indicated which were using similar methods.

Table 1 does not deliver any significant additional information other than that certain coral species were dominant. Also, it seems that numbers of coral colonies have been pooled between both 2005/2006 and 2010/2011 surveys. I suggest either a better description on where these numbers come from (just one or multiple surveys?) or delete this table completely. Above, I have suggested a potential justification to study these 5 coral species based on their dominance.

Table 2: There seems to be an error in Line 498 indicating the number 60? Please explain this number.

Figure 1 could be used as a justification to investigate the 5 corals of the study if you also indicate ‘other’ corals. You can argue that ‘other’ corals represent a negligible part of the reef of e.g. <10% or <5%. This way, table 1 can be excluded from the manuscript since it wouldn’t contribute with any new information. The colors in Figure 1 are chosen a little bit too screaming for the eyes of the moderate reader.

In Figure 2, you compare the difference in temperature between 2005 and 2010. However, temperature values for both 2005 and 2010 are only available from June 2005. Values for only 2010 can be excluded from this Figure since they don’t allow for comparison. Same accounts for the DHW figure. Please name here and elsewhere the panels in each figure as a), b), c),… to comply with the journal's criteria considering figures.

Figure 5 and Figure 6: Please indicate for which panels this accounts: ‘Lack of symbol in some panels indicates that no colonies were sampled for that specific coral at that given site during that specific survey period’ in the figure description. If possible, use similar graphical representation between Fig. 5 and Fig. 6 – the narrow panels in Fig 6 are distracting to the wider panels in Fig. 5.

Discussion:
Please start the discussion at Line 191 and develop you chain of arguments from the more specific to the more general findings and implications of you study. Line 185 – 191 are important but deviate from the study and can be excluded completely. Alternatively, you can significantly shorten this part and add it at the very end when you describe the implications of this study for conservation.

Line 286 – 292: Importantly, regions with exposure to a naturally high water flow (Bayraktarov et al, 2013, Nakamura et al, 2003, West and Salm 2003, Nakamura and van Woesik, 2001) can also represent refuge habitats for corals against bleaching.

Line 295 – 302 need to be moved into the Introduction and description of study site.

Line 306: Similar behavior is also seen for coral reefs dwelling in upwelling regions where the upwelling water masses supply reefs with increased nutrients and plankton.

---

## Round 0.2 · accepted · Accept

Both reviewers (and myself) feel that you have revised the manuscript well, and it is now acceptable for publication. I look forward to seeing this work in its published form.

·

Basic reporting

Looks good now.

Experimental design

No concerns.

Validity of the findings

Very interesting findings.

Additional comments

Well done.

·

Basic reporting

The manuscript has been significantly improved and has now a much clearer focus on the potential of transient turbid water masses to reduce coral bleaching. I am happy to recommend this manuscript for publication.

Experimental design

The authors have done a great job in clarifying their experimental design.

Validity of the findings

Findings are clear and valid.

Additional comments

No comments